# Host Transcriptome and Microbial Variation in Relation to Visceral Hyperalgesia [note 1]

**DOI:** 10.3390/nu17050921

**Published:** 2025-03-06

**Authors:** Christopher J. Costa, Stephanie Prescott, Nicolaas H. Fourie, Sarah K. Abey, LeeAnne B. Sherwin, Bridgett Rahim-Williams, Paule V. Joseph, Hugo Posada-Quintero, Rebecca K. Hoffman, Wendy A. Henderson

**Affiliations:** 1Department of Medicine, UConn Health, 263 Farmington Ave, Farmington, CT 06030, USA; chrcosta@uchc.edu; 2Inova Health Services, L.J. Murphy Children’s Hospital, 3300 Gallows Rd, Falls Church, VA 22042, USA; stephanie.prescott@inova.org; 3National Institute of Nursing Research, National Institutes of Health, 31 Center Drive, Bethesda, MD 20892, USA; 4Laboratory of Neuroimaging, National Institute of Alcohol Abuse and Alcoholism, 10 Center Drive, Bethesda, MD 20814, USA; 5Sinclair School of Nursing, University of Missouri System, 915 Hitt Street, Columbia, MO 65203, USA; sherwinl@missouri.edu; 6Office of Research and Sponsored Programs, University of North Florida, 1 UNF Drive, Jacksonville, FL 32224, USA; bridgett.rahimwilliams@unf.edu; 7National Institute on Alcohol Abuse and Alcoholism, National Institute on Deafness and Other Communication Disorders, National Institutes of Health, 10 Center Drive, Bethesda, MD 20814, USA; paule.joseph@nih.gov; 8Department of Biomedical Engineering, University of Connecticut, 260 Glenbrook Road, Storrs, CT 06269, USA; hugo.posada-quintero@uconn.edu; 9Laboratory of Innovative and Translational Nursing Research, School of Nursing, University of PA, 418 Curie Blvd, Philadelphia, PA 19104, USA; rhoffman@upenn.edu; 10Department of Biobehavioral Health Sciences, School of Nursing, University of PA, 418 Curie Blvd, Philadelphia, PA 19104, USA

**Keywords:** microbiome, oral, visceral hyperalgesia, gene expression, abdominal pain, human microbiome [D064307], visceral pain [D064307], abdominal pain [D015746]

## Abstract

Background: Chronic visceral hypersensitivity is associated with an overstressed pain response to noxious stimuli (hyperalgesia). Microbiota are active modulators of host biology and are implicated in the etiology of visceral hypersensitivity. Objectives: we studied the association between the circulating mRNA transcriptome, the intensity of induced visceral pain (IVP), and variation in the oral microbiome among participants with and without baseline visceral hypersensitivity. Methods: Transcriptomic profiles and microbial abundance were correlated with IVP intensity. Host mRNA and microbes associated with IVP were explored, linking variation in the microbiome to host RNA biology. Results: 259 OTUs were found to be associated with IVP through correlation to differential expression of 471 genes in molecular pathways related to inflammation and neural mechanisms, including Rho and PI3K/AKT pathways. The bacterial families Lachnospiraceae, Prevotellaceae, and Veillonellaceae showed the highest degree of association. Oral microbial profiles with reduced diversity were characteristic of participants with visceral hypersensitivity. Conclusions: Our results suggest that the oral microbiome may be involved in systemic immune and inflammatory effects and play a role in nervous system and stem cell pathways. The interactions between visceral hypersensitivity, differentially expressed molecular pathways, and microbiota described here provide a framework for further work exploring the relationship between host and microbiome.

## 1. Introduction

The neurophysiologic etiology underlying visceral hypersensitivity remains poorly understood but is thought to involve interaction between oral intake, microbial populations, and psychologic underpinnings. Visceral pain or hypersensitivity is a symptom that affects a large number of people and is intimately associated with gastrointestinal (GI) disorders and conditions [1]. Visceral hypersensitivity is a condition characterized by abnormal pain in response to typical non-painful stimuli, such as small molecular carbohydrate luminal content (allodynia), or by an exaggerated visceral pain response to noxious stimuli, such as mild intestinal cramping interpreted as intense acute abdominal pain (hyperalgesia). Persons suffering from visceral hypersensitivity may experience acute or chronic symptoms (or both), which may be intermittent or continuous. Hypersensitivity and chronic pain may be caused by peripheral and/or central sensitization of nociceptive neurons, often experienced in different ways than somatic sensation [2,3,4]. These neurons can be primed and activated by inflammatory mediators, and this hyperactivity may persist after inflammation resolves [5,6,7]. Both afferent and efferent neural networks support brain–gut communication, and recent research has highlighted the importance of the human microbiota as a mediator of this communication [8,9]. Aberrations in this communication network play a role in both organic and non-organic GI disorders, such as irritable bowel syndrome (IBS) [10,11].

IBS serves as a model for studying chronic visceral hypersensitivity and altered bowel motility due to its characteristic symptoms, including altered bowel motility, in the absence of gross inflammation, anatomical pathology, or acute infection. Successful treatment remains notoriously inconsistent [12]. The microbiota interacts with and regulates many physiologic and metabolic pathways, including gut motility, total body metabolic processes, and neurophysiologic axes, with cross-talk between microbes and host cells of the immune and nervous system being central to their effects [13,14,15,16]. Bioactive small molecules, both ingested and produced by resident microbes, may contribute to the experience of visceral hypersensitivity [17,18,19,20]. In lab animals, disruption of native microbiota via antibiotic treatment changes the expression of pain channels, even on distal dorsal root ganglia, suggesting a role for microbiome composition in hypersensitivity [21].

Traditional methodologies for studying microbial community variation in GI disorders have focused on luminal microbiota (feces) or colonic mucosal microbiota. There is noted overlap in the composition of the intestinal and oral microbiome in both physiologic and pathologic states (e.g., colorectal cancer, IBS, and obesity) [22,23]. Contemporaneous sequencing of oral mucosa-adherent bacteria and intestinal microbes have distinct population shifts in both communities, driven by changes in Bacteroides abundance in the intestines and Fusobacteria in the oral cavity [23]. However, these shifts differed between individuals with active and quiescent IBS and healthy controls. Other groups have shown shifts in Bacteroides populations in the oral cavity as well [24,25,26].

In the oral cavity, this interaction has come under particular focus in recent investigations. Oral mucosa-adherent microbial profiling has recently become a focus of attention. Differences in the oral microbiome of healthy controls and patients for a number of disease types have been described, most notably diseases with an inflammatory component, such as atherosclerosis, inflammatory bowel disease, IBS, and rheumatoid arthritis [27,28,29,30,31,32]. The microbial composition of the oral cavity and its impact on overall dental health has been shown to impact neuronal functioning. Work carried out by several groups has identified shifts in periodontal microbe populations (specifically enrichment of *Prevotella*) in individuals with APOE-4 mutations associated with dementia [33]. Additionally, poor dental health and pathologic changes of the resident microbes has been linked to increased risk of cognitive impairment with aging [34].

Oral microbiome changes or infection may lead to systemic insult from bacterial toxins, systemic low-level inflammation and immune response, and/or the translocation of oral pathogens to distant tissues (Figure 1A) [28]. Comparison of the microbial communities across various body sites, inclusive of the oral cavity and gut, indicates that microbial community types present in the mouth and gut are predictive of one another, though the same bacteria do not reside in both locations. The oral microbiome may predict intestinal dysbiosis [35]. Thus, GI conditions and systemic symptoms associated with IBS and inflammatory bowel disease (IBD), among others, may be associated with oral microbiome alterations. Recent research provides tentative support for this hypothesis (Figure 1A), as IBD patients show reduced oral microbiome diversity compared to healthy controls, paralleling the loss of diversity previously described in the intestinal microbiome [27,30]. In this study, we examined whether the abundance of oral mucosa-adherent microbes correlated to the severity of biochemically induced visceral pain (IVP) in a cohort of participants with and without chronic visceral hypersensitivity baseline. We investigated the correlation between the peripheral global transcriptome (i.e., mRNA isolated from blood samples) and the microbiome for potential molecular pathways that may provide clues to the functional links between microbial imbalance, systemic host biology, and IVP severity or clinical phenotypes (i.e., IBS–C, IBS-D, and IBS-M) (Figure 1B). The investigation of specific microbial associations by exploiting IBS-associated hyperalgesia (i.e., induced visceral pain (IVP) severity in response to a noxious stimulus) will not only help identify novel diagnostic microbial markers but may also suggest candidate molecular pathways underling visceral hypersensitivity.

## 2. Materials and Methods

### 2.1. Participants

Both healthy controls (HCs) (n = 20) and participants with irritable bowel syndrome (IBS) (n = 20) were carefully screened, phenotyped, and matched for race, sex, age, and weight; participants in the age range 13–45 years old were eligible for inclusion (females were required to have menses for at least 2 years) (Table 1) [36,37]. Participants with IBS were diagnosed according to Rome III criteria (symptom duration of at least 6 months, with 3 or more symptom days per month for 3 months and two of three additional symptoms: improved pain with defecation, change in stool appearance, and change in stool frequency in absence of other disease process) [38]. All participants signed a written consent form to have their samples stored and used for research purposes at the National Institutes of Health (NIH) Hatfield Clinical Research Center, Bethesda, MD, USA. Exclusion criteria included the following: history of organic bowel disease (e.g., IBD, colorectal cancer), use of laxatives or other prokinetic medications (fiber supplementation was permitted), severe psychiatric or co-morbid pain syndrome, night-shift workers, or excessive caffeine (>300 mg) or alcohol consumption (more than 2 standard drinks daily). The study was approved by the National Institutes of Health Institution Review Board, NCT00824941. Full inclusion and exclusion criteria are available at clinicaltrials.gov using the referenced trial number.

### 2.2. Experimental GI Stressor

Induced visceral pain (IVP) was measured in response to an orally administered simple sugar test solution by using a Gastrointestinal Pain Pointer (GIPP) instrument (PST, Inc., Pittsburgh, PA, USA [36,37]. The sugars composing the test solution (mannitol, lactulose, sucralose, and sucrose) have been shown to permeate the epithelial border [36]. Among IBS patients, increased permeability is associated with increased visceral hypersensitivity, and use of this assay was intended to exacerbate the visceral hypersensitivity response [39]. The maximum IVP scores observed and recorded were utilized for downstream analyses.

### 2.3. Microbiome Profiling

Buccal swabs were collected using a Cytobrush^®^ (CooperSurgical, Berlin, Germany). Buccal cells and mucus were mechanically separated in phosphate-buffered saline, pelleted, and stored at −80 °C until extraction. DNA was extracted using BiOstic^®^ Bacteremia DNA Isolation Kits (MO-BIO Laboratories, Carlsbad, CA, USA). The manufacturer’s instructions were followed without modification.

The extracted DNA was shipped to SecondGenome for amplification, purification, hybridization, and microarray analysis. SecondGenome processed the samples in a “Good Laboratory Practices”-compliant laboratory. Bacterial 16S rRNA was amplified by polymerase chain reaction (PCR) using the 27F.1 forward primer and 1492R.jgi reverse primer. Thirty-five cycles of PCR amplification were performed. The amplified PCR product of each sample was purified using a solid-phase reversible immobilization method. The purified PCR products were quantified using an Agilent 2100 Bioanalyzer^®^ Santa Clara, CA, USA Thirty-nine samples passed the PCR quality control (QC) criteria and were moved forward to hybridization. Bacterial 16S rRNA gene amplicons were fragmented, biotin labeled, and hybridized to the PhyloChip™ Array, version G3. PhyloChip arrays were washed, stained, and scanned using a GeneArray^®^ scanner (Affymetrix). Each scan was captured using standard Affymetrix software (GeneChip^®^ Microarray Analysis Suite, Santa Clara, CA, USA). The scan for one sample did not pass required QC specifications and was excluded from data analysis. Data for 38 samples were moved forward to data analysis. One sample was found to be an extreme outlier, and another did not yield usable gene expression data. A total of 36 samples (18 HC and 18 IBS) were used in downstream analyses. The microbiome data that support the findings in this study are available from the corresponding author, WAH, upon reasonable request.

### 2.4. Whole Genome Gene Expression

Whole fasting blood samples were collected via venipuncture in PAXGene tubes (PreAnalytiX, Qiagen, Valencia, CA, USA) as per the manufacturer’s instructions and stored at −80 °C until RNA purification. Total RNA was extracted using the PAXGene Blood miRNA extraction kit (PreAnalytiX, Qiagen, Valencia, CA, USA), and RNA quality was evaluated on a 2100 Bioanalyzer (Agilent Technologies, Palo Alto, CA, USA) and a NanoDrop™ 1000 spectrophotometer (Wilmington, DE, USA). All RNA samples had RIN numbers greater than 7. RNA samples were stored at −80 °C. Samples were de-identified as per the approved provisions of the protocol.

In vitro transcription products were prepared from 10 ng of total RNA at the Laboratory of Molecular Technology, National Cancer Institute (NIH, Frederick, MD, USA). Fragmented, amplified RNA was labeled with the Ovation Whole Blood Solution (NuGen Technologies, San Carlos, CA, USA). Four micrograms of cDNA was hybridized to Affymetrix GeneChip Human Genome U133 Plus 2.0 Arrays and washed and scanned according to the manufacturer’s instructions. Raw data were normalized by using the global mean method. Probe-set signal values were natural log transformed and scaled by adjusting the mean intensity to a target signal value of log 500. Unspecified probes were excluded from further analysis. Gene-specific probe set expressions were then summarized as a single gene-specific expression value. The sequencing data that support the findings of this study are openly available in the NCBI Gene Expression Omnibus at https://www.ncbi.nlm.nih.gov/geo, reference number GSE109597. Deposition date 24 January 2018. Last updated 18 June 2019.

### 2.5. Statistics

#### 2.5.1. Data Analysis of Microbial Abundance, Gene Expression, and IVP

To assess the relationships between microbial abundance, peripheral mRNA expression, and IVP severity, data were reduced by retaining operational taxonomic units (OTUs) and mRNAs that significantly correlated to the IVP score reported by participants. Generally, lack of visceral pain response was associated with HC group and presence of a visceral pain response was associated with IBS group. Correlation analysis largely reflected categorical comparisons. Therefore, only data from participants who experienced and reported IVP in response to the GI test solution were used to circumvent the possible bimodal effect of such data on correlation. Pearson’s correlation with Bonferroni correction for multiple comparisons was used for correlation of IVP and microbial OTUs, similar to methods described previously by our group [40].

The resulting IVP-correlated gene and microbial lists were then correlated with one another. These correlations were further examined by singling out gene expression correlations of dominant and relevant phylotypes. These gene lists were examined by looking at the magnitude of the correlations, as well as the number of correlations to a given gene across phylotypes, and vice versa. Gene lists were further interrogated using Ingenuity Pathway Analysis (IPA, Ingenuity Systems, Redwood City, CA, USA) to discover functional themes, pathways, interactions of interest, and disease associations.

#### 2.5.2. Data Analysis of Microbiome Profiles

The presence and absence of OTUs were used to characterize the profiles of different subpopulations within the sample, specifically comparing HC and IBS/IBS-subtypes. Each category was characterized independently; profiles were then compared among categories. To focus on meaningful changes in bacterial representation, the analysis was limited to OTUs with a substantial presence (≥30% of samples) within the group under consideration (i.e., the abundance of an OTU present in 30% of HC samples was compared to its abundance in IBS samples), which helps avoid the outlier effect of a single sample on the group’s microbiota composition. Abundances were considered significant if there was a greater than 25% difference in presence between HC and IBS groups. To further avoid the effects of single samples, distribution of higher order taxonomic classifications (e.g., phyla, family) were used for comparison, as many OTUs remain undescribed at the genus and species level. Finer-grained functional and/or ecological similarities may also be united at the family level but may be lost at higher taxonomic ranks, such as phyla or class. The rate of representation of individual OTUs was also analyzed. This approach was taken to identify specific taxa that are commonly or exclusively found in one group and not another, an approach developed from prior experience of analyzing oral microbiomes [40].

## 3. Results

### 3.1. Severity of Induced Visceral Pain (IVP)

A total of 11% of the HC reported IVP (IVP Score = 8.1 ± 2.7; range: 6.2–10 on a scale of 0–100), and 89% of the IBS participants reported IVP (IVP Score = 32.7 ± 26.7; range: 4–81) following the ingestion of the GI test solution (Figure 1B) [26].

### 3.2. Microbial and Transcriptome Correlations with Induced Visceral Pain (IVP)

A total of 259 OTUs were significantly correlated (r^2^ > 0.2, *p* < 0.05, prior to adjustment for multiple comparisons) to variations in the severity of IVP experienced across HC and IBS participants (Appendix A). Nearly a quarter of all OTUs, which were negatively correlated to IVP (194 OTUs), belonged to the family Lachnospiraceae, followed by Moraxellaceae (9%), Pseudomonadaceae and Rikenellaceae (6%), and Bacillaceae (5%). All other families (n = 30) accounted for less than 3% of the remaining OTUs. Similarly, the majority of OTUs that were positively correlated to IVP (65 OTUs) belonged to the family Lachnospiraceae (18%), followed by Prevotellaceae (12%), Veillonellaceae (11%), Porphyromonadaceae (8%), and Erythrobacteraceae (6%). The remaining families (n = 10) accounted for less than 3% of the OTUs positively correlated to IVP each (Appendix A).

Of the 259 OTUs that were correlated to IVP, seven correlations had *p*-values close to or lower than the Bonferroni corrected *p*-value (*p* = 0.00005) for multiple comparisons (Appendix A). These were all negatively correlated to IVP, with squared regression coefficients (r^2^) greater than 0.63 (range: 0.63–0.72). None of these bacteria had been described at the species level, although genus-level identification was available for three OTUs, which included two OTUs belonging to the genus Bacillus and one belonging to the genus Clostridium.

Correlation analysis identified 508 genes that were significantly correlated to IVP intensity (Appendix A). Correlations were generally weak to moderate (average r^2^ = 0.25, range: 0.19–0.63) and statistically significant (*p* < 0.05; range: *p* = 0.00002–0.05). None of the significance levels of the correlations passed the Bonferroni multiple comparisons corrected *p*-value (*p* = 0.000003). Canonical pathway analysis (Table 2) found that pathways relating to stem cell pluripotency and multiple sclerosis were significantly (*p* < 0.05) represented in the gene list (Appendix A).

### 3.3. Correlations Between IVP-Correlated Genes and Microbes

The 259 IVP-correlated OTUs were significantly correlated to at least 1 of 471 IVP-correlated genes (Appendix A). Of these, 196 OTUs were highly correlated to 130 genes (r^2^ = 0.5–0.81). OTUs from seven bacterial families dominated the correlations (number of gene correlations: Lachnospiraceae—395; Pseudomonacocaceae—258; Rikenellaceae—245; Moraxellaceae—182; Bacillaceae—163; Veillonellaceae—135; and Prevotellaceae—126). There was considerable overlap between each family’s correlation gene list. A total of 78 genes were highly correlated (i.e., r^2^ > 0.5) with at least 1 OTU from the 7 dominant bacterial families. Common canonical pathways and disease associations among all seven bacteria-associated gene lists, common gene lists, and high-correlation (i.e., > 50%) gene lists included multiple sclerosis, cytokine pathways, Rho signaling, and the PI3K/AKT pathway, suggesting a theme of inflammatory- and immune-related associations between bacterial profiles, host gene expression, and IVP (Figure 2A,B and Figure 3; Appendix A).

Four OTUs, identifiable to the species level, showed moderate, positive correlations with IVP severity (Campylobacter gracilis, Dialister invisus, Tannerella forsythia, and Veillonella dispar), and one (Mycoplasma hominis) was highly positively correlated (r > 0.7; Figure 4A). Several of these species were highly (>50%) correlated to genes expressed in peripheral circulation (CDHR1, CHRM2, GDF9, LAMC3, METTL21A, PACSIN21, TMEM231, TMEM50B, and UCA1) (Figure 4A).

Eight species showed moderate negative correlations to IVP severity (Brevibacterium paucivorans, Rothia aeria, Rothia dentocariosa, Capnocytophaga sputigena, Acinetobacter venetianus, Xanthomonas retroflexus, Sphingomonas echinoides, and Aerococcus viridans) (Figure 4B). Two species were highly (r < −0.7) negatively correlated with IVP severity (Gemella sanguinis and Acinetobacter johnsonii) (Figure 3 and Figure 4B). All but one of these species were highly (> 50%) correlated with peripherally expressed genes (APOC1, CACHD1, CCT6B, DNAH3, HAP1, MAP7D2, NDRG4, NPHP4, OR2B6, PLEKHA2, PROSC, SCARNA17, SORBS2, SP140L, SPTSSB, SRRM5, SUV39H2, TMED6, and ZFY) (Figure 3 and Figure 4).

### 3.4. Microbial Profiles Related to IBS and IBS-Subtypes

The Lachnospiraceae were the most dominantly represented family in both IBS (303 OTUs present in 30% or more samples) and HC (325 OTUs present in 30% or more samples) groups, accounting for approximately 21% of all bacterial OTUs present in either group. Overall, proportional representation of different bacterial families did not differ by more than 2% between groups for any of the families compared (Appendix A).

Of the OTUs, 27 were ≥25% more common in IBS participants than HCs (i.e., these OTUs were present in 25% or more IBS individuals than in HCs) (Table 3 and Appendix A), while 41 OTUs were ≥25% more common in HCs than IBS participants (Table 4). Of the OTUs commonly expressed in IBS but not in HC participants, seven (22%) belonged to the family Lachnospiraceae but were not identified to the genus or species level, and three (11%) belonged to the family Prevotellaceae (unspecified *Prevotella* sp.). Four OTUs were either completely absent or 100% present in at least one patient cohort (Table 3). One OTU belonging to the family Rikenellaceae (genus and species unspecified) and one to the family Desulfobacteraceae (genus and species un-specified) were present in all IBS-C participants; these same OTUs were present in less than 75% of IBS-D cases and less than approximately 50% of HCs. One Fusobacteriaceae OTU (unspecified *Fusobacterium* sp.) was always absent in HC, and another unspecified Fusobacterium species was always absent in the IBS-D participants (Table 3).

IBS-mixed participants were not included because this group only consisted of two individuals. Of the 41 OTUs commonly found in HCs but not in IBS participants (Table 4), 17 (41%) belonged to the Lachnospiraceae family (all of unspecified genus and species except for 1 unspecified *Coprococcus* sp.), and 4 (10%) belonged to the Pseudomonadaceae family (unspecified *Pseudomonas* sp.). Overall, of these OTUs, 14 (34%) were either 100% present in HCs or 100% absent in IBS or IBS-subtypes. One Lachnospiraceae OTU of unspecified genus and species was present in 100% of HCs but only present in 72% of IBS participants; with particularly low representation in IBS-D participants (57%) compared to IBS-C participants (88%). A total of 1 unspecified Pseudomonas species (family: Pseudomonadaceae) was present in all 18 HCs and all 9 IBS-C participants, but it was only present in 43% of IBS-D participants. Three unspecified Lachnospiraceae OTUs and one unspecified OTU classified within the phylum Verrucomicrobia were always absent in IBS participants but substantially present in HCs. A total of 9 OTUs (family: Lachnospiraceae × 7; phylum: Proteobacteria × 1; phylum: Verrucomicrobia × 1) were absent in 100% of IBS-D participants. A total of 7 OTUs (family: Coriobacteriaceae × 1; family: Lacnospiraceae × 5; phylum: Verrucomicrobia × 1) were absent in all IBS-C participants.

Comparison of the abundance (i.e., expression) of 1049 bacterial OTUs showed that 86 OTUs were significantly differentially represented, although none passed multiple comparison correction of the alpha value (Bonferroni corrected alpha = 0.00005). Of these, 65 OTUs (representing 13 families, with 38% of UTOs belonging to the Lachnospiraceae) were more abundant, and 21 OTUs (7 families; 42% Prevotellaceae) were less abundant in IBS participants relative to HCs (Table 4). Comparison of bacterial abundance in IBS-D participants to HCs revealed 82 OTUs that were significantly differentially represented, although none of the alphas passed the Bonferroni correction. Of the OTUs, 66 (25 families, 27% Lachnospiraceae) OTUs were down-regulated and 15 (10 families) were up-regulated in IBS-D participants compared to HC. IBS-C participants differed from HC in the expression of 28 bacterial OTUs; however, none passed multiple correction criteria. Finally, 16 (7 families) were down-regulated and 12 (6 families) were up-regulated in IBS-C compared to HC. Sixteen host genes associated with butyrate were correlated with both IVP and Lachnospiraceae expression (Figure 5). Matrix and functional linkage network analysis was performed to explore the relationship of neuronal pathways and function as related to IVP (Figure 4B; Appendix A).

## 4. Discussion

Using microarray-based technology, we characterized the oral mucosa-adherent microbiome in participants with and without chronic visceral hypersensitivity. We assessed both the microbiome abundance and diversity (i.e., the presence and absence of specific OTUs). We further determined the global transcriptome profiles of each participant. Both mRNA levels and microbiome abundance data were analyzed to examine their relationships to an induced hyperalgesic response (i.e., intensity of IVP) across all participants who reported a response to the test solution. Our analysis revealed robust associations among several known bacteria species, genera, families, and IVP severity. Specifically, robust correlations between the abundance of specific bacteria and the severity of IVP, as well as novel associations between pain-associated bacterial abundance and pain-associated peripheral mRNA expression levels in participants who responded to the test solution, were observed. These associations suggest a system-level dysregulation of nervous system- and inflammation-related pathways. Finally, we compared oral microbial profiles using the clinically defined groups (HC, IBS, IBS-D, and IBS-C). This approach yielded evidence of a differential presence of certain bacterial types and families in IBS and IBS-subtypes. These results were consistent with the existing IBS-microbiome literature.

We found significant associations among bacteria, host mRNA expression patterns, and induced visceral pain (Figure 2, Figure 3 and Figure 4), with new associations between bacteria linked to pain and peripheral mRNA levels related to inflammation and nervous system function. These findings suggest systemic dysregulation in inflammatory pathways (e.g., cytokine networks, PI3K, and JAK/STAT) and neurological pathways (e.g., BDNF and CAV2) detectable in peripheral blood, highlighting the potential systemic basis of visceral hypersensitivity. The fact that these correlations were found with genes expressed in peripheral blood circulation suggests that symptom-related inflammatory (e.g., cytokine networks, PI3K, JAK/STAT, and AKT pathways) and neurological (e.g., BDNF and CAV2) dysregulation is peripherally detectable and that the etiology of the condition/symptom may have a systemic basis. A closer look at the specific bacterial families and their metabolic products may provide clues to this systemic etiology.

The family Lachnospiraceae was particularly interesting because its members showed significant variation in diversity and abundance in patients who experienced IVP and among clinically defined groups [41,42]. Lachnospiraceae is a family of bacteria which is particularly diverse and abundant in the digestive tract of mammals [43,44]. The family also contains numerous short-chain fatty acid (SCFA)-producing members, especially microbes which produce butyric acid [45,46,47]. Butyric acid is an end product of sugar and starch fermentation and has a suppressive effect on some microbes, while being a principal source of energy to other microbes and an essential energy source within the host epithelium [15,18,48,49,50,51,52]. It also has protective anti-inflammatory and anticancer effects [48,53]. Sixteen host genes associated with butyrate were correlated with both IVP and Lachnospiraceae expression (Figure 5. The decreased diversity (Table 4) and abundance (Appendix A) of this family of bacteria in IBS patients, particularly IBS-D patients, suggests that butyrate production may be affected. Others have reported decreased expression in butyric acid-producing bacteria and phylotypes, all of which lends support to our oral-mucosa-derived data; however, further work needs to be carried out to correlate our findings to luminal microbiome composition [54,55]. It should be noted, however, that the Lachnospiraceae is a large family with functional diversity, and although most members of this family are negatively correlated with IVP, some members are positively associated. Further functional characterization at a species level is needed to better understand the role these microbes play.

In contrast to the down-regulation of Lachnospiraceae, the family Prevotellaceae, and specifically the genus Prevotella, showed increased abundance in IBS patients. Prevotella have been found to be especially dominant in the GI tract of individuals with high-carbohydrate, low-protein diets [56]. This genus is also a common pathogen in periodontal disease, which has been associated with a number of non-oral/non-GI pathologies, such as cardiovascular disease and arthritis, and may have pro-inflammatory properties and be especially well-adapted to inflamed environments [57,58,59]. Prevotella was also found to have an increased oral abundance in inflammatory bowel disease (IBD) patients and was positively associated with the inflammatory cytokine IL1β levels in these patients [30]. Elevated abundance of Prevotella and Veillonella has been described in pediatric IBS patients, particularly pediatric IBS-C patients [60]. Similar elevated GI abundance of Prevotella has also been described in celiac disease, diabetes, and colorectal cancer [61,62,63]. Our results show an association between an increase in oral Prevotellaceae (*Prevotella* sp.) and Veillonellaceae diversity and abundance and IVP severity. These results are broadly convergent with previous findings and support growing evidence of a subclinical inflammatory component of IBS.

Furthermore, our results show that several members of the family Veillonellaceae are significantly more highly expressed in IBS-D patients (Appendix A) and are positively correlated with IVP severity (2 × Dialister invisus sp., Dialister pneumosintes, 2 × Veillonella sp., and Veillonella dispar) (Figure 4A). This family also appears to be more diverse in IBS, and specifically IBS-D, patients compared to HCs (Appendix A). Several studies have found that, specifically, the Veillonellaceae genus Veillonella tends to be over-represented in IBS patients [64]. Both Veillonella sp. and Dialister sp. produce acetic, propionic, and succinic acids as metabolic end products [65]. These acids can regulate a number of genes that show correlation with Veillonellaceae abundances. Although regulation is mostly indirect, these genes also form part of molecular inflammatory pathways. Overgrowth in Prevotellaceae and Veillonellaceae appear to be biomarkers of the non-normal physiologies of patients with GI symptoms, and as our data show, could be good oral biomarkers of more systemic microbial dysregulation and inflammation (e.g., CCR5, AKT3, and SOCS4—see Figure 2 and Appendix A) through the biological effects of their metabolic end products on epithelial biology, as well as systemic processes.

Gene lists, functional networks, and canonical pathways suggest an underlying systemic theme relating to inflammation, neurological process, and stem cell pluripotency (Canonical Pathways Analysis—Table 2). OCT4 and NANOG stem cell signaling relate to genes such as the growth-associated protein 43 (GAP43), which plays a vital role in regulating the plasticity of the ENS, in which stem cells need to migrate to respond to changes in the microenvironment [66,67]. GAP43 is essential for maintenance of the axon and neuronal functioning. A significant amount of cross-talk has been shown to exist between the resident macrophages in the intestinal wall and ENS cells, and previous work has shown that gut motility can be altered via bone-derived morphogenic protein signaling from macrophages to ENS neurons. Gut microbiota can induce production of this protein in macrophages, and antibiotic treatment can alter this response, demonstrating the inter-relatedness of microbiota, immune cells, and the ENS [68].

Overall, our results suggest immune, inflammatory, and nervous system mechanisms contribute to the etiology of chronic visceral hypersensitivity. By connecting microbiome data and the global transcriptome profiles with each other through the quantifiable severity of visceral hyperalgesia, we began to associate variations in the microbiome with functional domains of the host biology. Although our sample size is relatively small, our sample was carefully phenotyped and rigorously screened. The main strength of our analysis remains the fact that we did not rely on binary comparisons, which can severely underestimate the complexity of natural systems, but evaluated symptom severity across categories that yielded robust associations that will inform future research. We identified differential gene expression of molecular pathways linked to nervous system function, inflammation, and stem cell pluripotency in patients with visceral hypersensitivity. These may become important targets for future studies; however, we acknowledge that caution must be exercised in interpretation of these results as the systems observed are very complex. Additionally, given the cross-sectional design and relatively small sample size of our study, further work will need to be carried out to determine if these patterns persist among the population as a whole. Other limitations of our work include a lack of contemporaneous gut microbiome sequencing, limiting the assessment of broader host-microbe changes in both IBS and healthy controls. Future research needs to focus more closely on discovering the functional links between these genes/pathways and microbes and identify mechanisms through which the microbiome may modulate or be modulated by these genes/pathways in the context of chronic hypersensitivity, since functional assays were not included in our study. However, in the meantime, our phenotype data show that it may be possible to move toward interventions by manipulating the richness and abundance of the bacterial OTU identified in order to alleviate visceral hypersensitivity. Specifically, the stereotyped increases in Veillonellaceae with reduction in Lachnospiraceae and Prevotella oral populations may serve as a diagnostic signature of IVP with further study and validation.

This research provides a strong foundation for future research investigating the role of the microbiome in visceral hyperalgesia. In future studies, we will explore a continuous objective measure of pain based on electrophysiological signals that would greatly enhance these efforts [69]. By providing a more accurate and consistent measure of pain intensity, an objective pain index may offer added evidence of their pain experience, potentially identifying subgroups with distinct microbial profiles and treatment responses. It may also improve treatment monitoring by tracking the effectiveness of interventions aimed at modulating the microbiome or targeting identified pathways, such as probiotics, prebiotics, or dietary changes, and enhance biomarker discovery.

## 5. Conclusions

Our work suggests a bidirectional relationship between microbiota and host genes, mediated primarily by the Lachnospiracae family, may underlie visceral hypersensitivity. Future research can focus on elucidating the functional connections between these genes and microbes, particularly to understand how microbiome modulation could influence gene expression and alleviate symptoms. Targeting the richness and abundance of bacterial OTUs identified here may represent a novel approach to managing visceral hypersensitivity.

## Figures and Tables

**Figure 1 nutrients-17-00921-f001:**
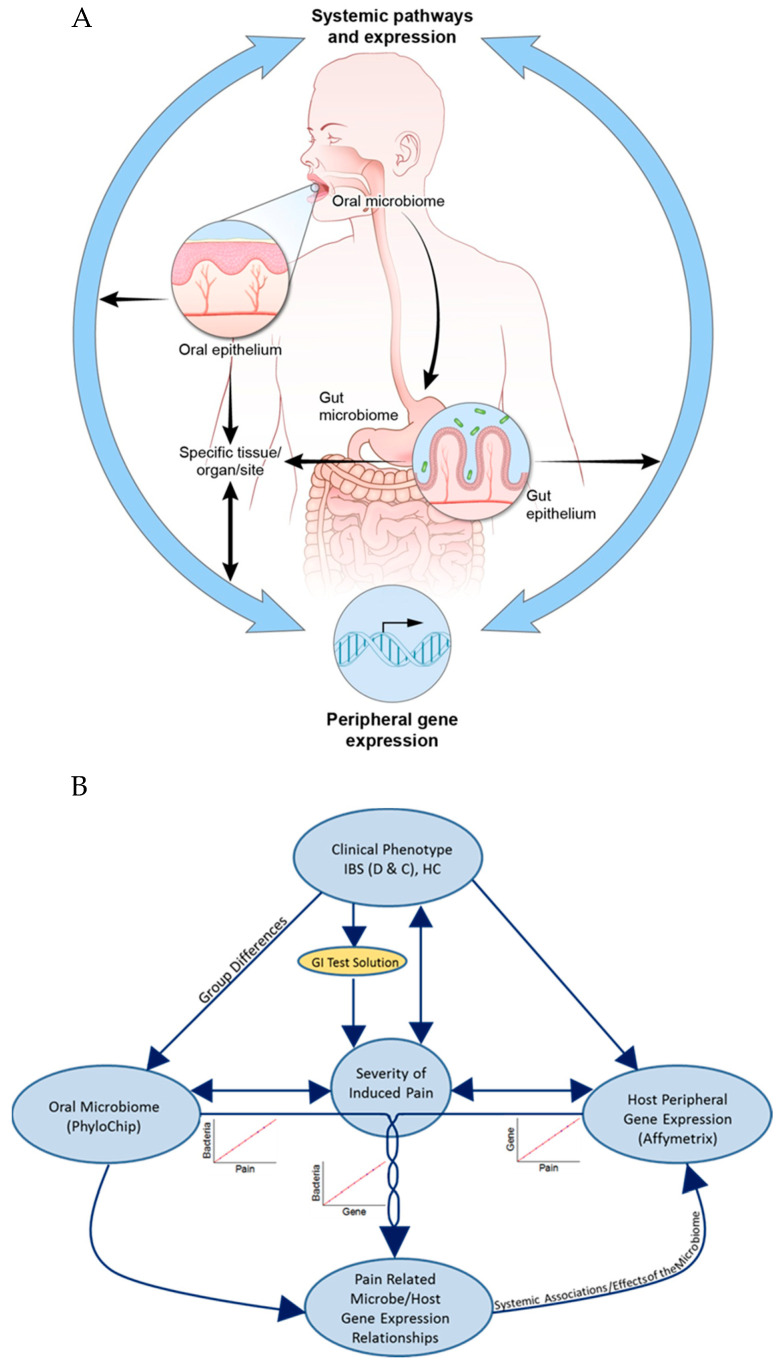
(**A**) Model of host transcriptome and microbial variation in relation to visceral hyperalgesia and its links to specific and systemic dysregulation. The oral microbiome may directly lead to or mediate gut microbiome changes within the lower GI tract and also be a direct source for the bacterial colonization of specific systems/sites/tissues/organs and induce both specific and systemic effects on the host biology, leading to visceral hypersensitivity and hyperalgesia. (**B**) Protocol design diagram. Visceral pain was induced in clinically phenotyped participants with the anticipation that a range in the severity of the induced pain would be elicited. Individual sensitivity (and, therefore, the severity of the induced visceral pain) to the GI stressor is speculated to have bidirectional relationships with the dysregulation of biological processes (characterized through gene expression data) and the microbiome, which, in turn, mediate each other.

**Figure 2 nutrients-17-00921-f002:**
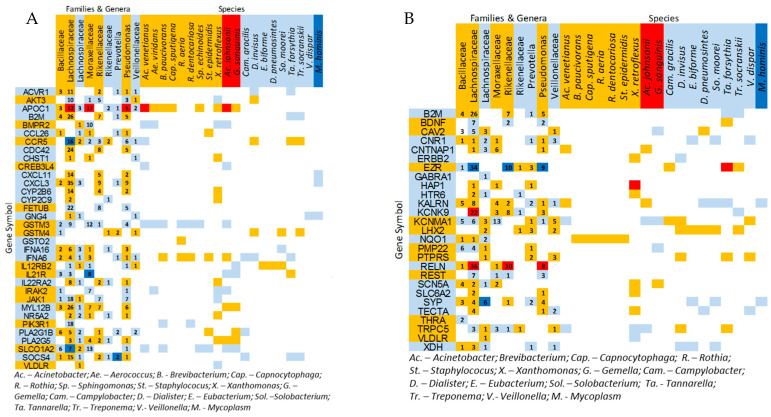
Matrices of bacterial OTUs and host genes as mediated by correlation to IVP. (**A**) Matrix of bacterial-associated genes related to cytokine signaling. All genes in the network correlated significantly with the severity of induced visceral pain (IVP). (**B**) Matrix of genes related to neuronal pathways and function and relationship to microbiome constituents. All genes in the network correlate significantly to the severity of induced visceral pain (IVP). Both tables are structured as follows: gene–IVP correlations (vertical bar), bacteria–IVP correlations (horizontal bar), and bacteria–gene correlations (colored cells). The number of OTUs belonging to each of several bacterial families and genera that correlate to a specific gene in the network is indicated by the number in each cell. Orange indicates negative correlations, and blue indicates positive correlations. Dark blue and red indicate that one or more of the correlations represented in a cell exceed 50%.

**Figure 3 nutrients-17-00921-f003:**
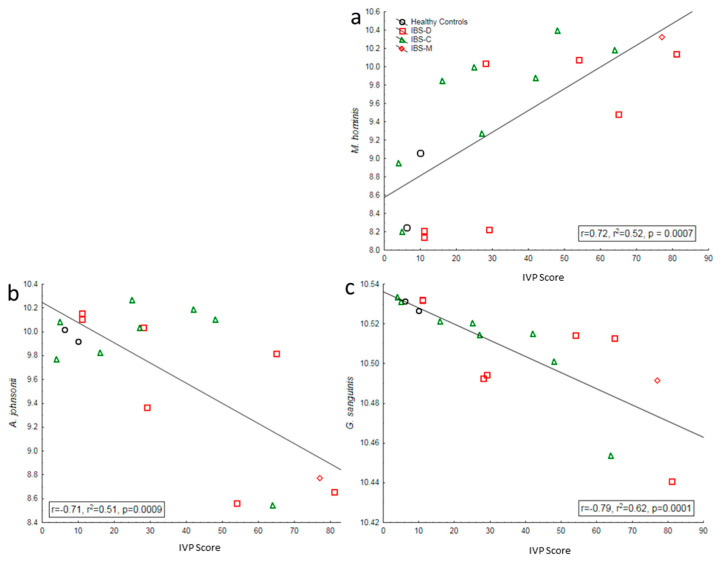
Correlation of bacterial abundance to IVP score. Significant correlations between the abundance of specific bacterial species (*y*-axis) with the IVP scores (*x*-axis). (**a**) Pearson correlation between M. hominis and IVP score which shows a significant positive correlation between bacterial abundance and IVP intensity. (**b**) Negative correlation between A. johnsonii abundance and IVP score. (**c**) Strong negative correlation between G. sanguinis abundance and IVP intensity.

**Figure 4 nutrients-17-00921-f004:**
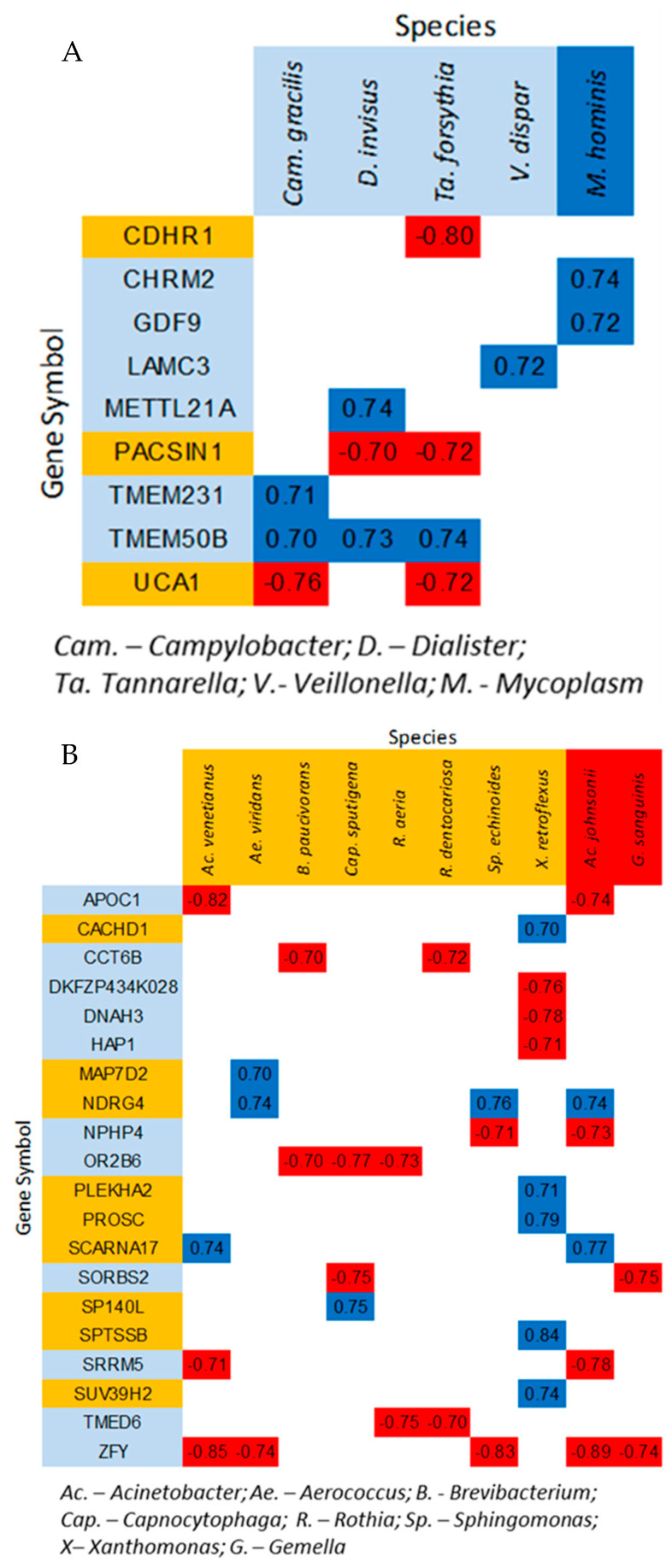
Correlation plots of bacterial abundances and IVP severity. (**A**) Correlation between specific bacterial species whose abundance increases with IVP severity and genes that are both positively and negatively associated with IVP severity (**B**) Correlation between specific bacterial species whose abundance decreases with IVP severity and genes that are both positively and negatively associated with IVP severity. For both A and B, positive correlations are highlighted in blue, and negative correlations are highlighted in orange and red. The color in which genes and bacteria are highlighted indicates the direction of their association with IVP. Dark blue and red indicate that the correlation exceeds or equals 50%.

**Figure 5 nutrients-17-00921-f005:**
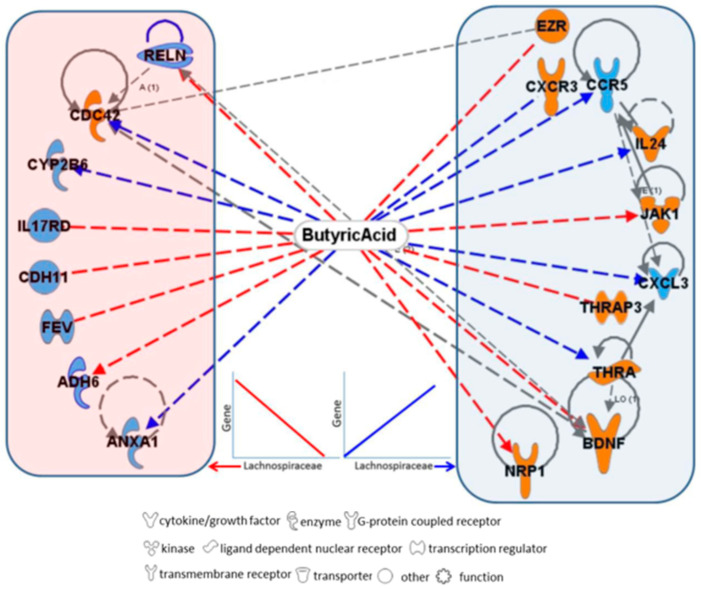
Functional pathway linkage network of induced visceral pain (IVP) with correlated mRNAs. These networks are also correlated significantly to Lachnospiraceae expression and are regulated by butyrate. Genes in orange are negatively correlated to IVP, and genes in blue are positively correlated to IVP. Red lines connecting genes with butyrate indicate that butyrate down-regulates the gene expression, while blue lines indicate that butyrate up-regulates its expression. Grey lines indicate indirect (dotted lines) and direct (solid lines) relationships between genes. Genes grouped in the blue box on the right are positively correlated to Lachnospiraceae expression, while genes grouped in the orange box on the left are negatively correlated to Lachnospiraceae expression.

**Table 1 nutrients-17-00921-t001:** Demographic, clinical data, and IVP scores for each study participant. The mean and standard deviation for age and induced visceral pain (IVP) for each group is provided. The proportions of the presence of demographic subgroups in the study population are given for sex and race.

	Chronic Visceral Hypersensitivity						Healthy Controls				
	Sex	Age	Race	Condition Subtype	IVP		Sex	Age	Race	Condition Subtype	IVP
n = 18	Male	44	Caucasian	Diarrhea	81	n = 18	Female	30	Caucasian	n/a	10
	Male	15	Caucasian	Mixed	77		Female	23	Mixed/Other	n/a	6.2
	Female	30	African-American	Diarrhea	65		Female	24	African-American	n/a	0
	Male	45	African-American	Constipation	64		Male	23	Caucasian	n/a	0
	Male	26	African-American	Diarrhea	54		Male	40	Caucasian	n/a	0
	Female	32	African-American	Constipation	48		Male	29	Asian	n/a	0
	Female	26	African-American	Constipation	42		Female	33	African-American	n/a	0
	Female	23	African-American	Diarrhea	29		Female	24	Caucasian	n/a	0
	Female	27	Caucasian	Diarrhea	28		Female	24	Caucasian	n/a	0
	Female	29	African-American	Constipation	27		Male	24	African-American	n/a	0
	Female	24	Caucasian	Diarrhea	25		Male	31	Caucasian	n/a	0
	Male	30	Caucasian	Constipation	16		Female	37	African-American	n/a	0
	Female	26	Asian	Diarrhea	11		Female	21	African-American	n/a	0
	Female	24	Caucasian	Diarrhea	11		Female	21	Asian	n/a	0
	Female	24	Caucasian	Constipation	5		Female	16	Mixed/Other	n/a	0
	Male	26	Caucasian	Constipation	4		Female	43	Mixed/Other	n/a	0
	Male	31	Asian	Mixed	0		Male	22	Caucasian	n/a	0
	Female	24	Mixed/Other	Constipation	0		Female	28	African-American	n/a	0
Mean	44% male	28.11	44% Caucasian		32.61	Mean	33% male	27.39	38% Caucasian		0.90
SD		7.09	38% African-American		26.76	SD		7.17	33% African-American		2.70

**Table 2 nutrients-17-00921-t002:** Results of canonical pathway gene analysis. Ingenuity Pathway Analysis was used to analyze pathways present in gene profiles of participants. Phosphorylation data were not included, so only pathways with differential expression were highlighted. Canonical pathway analysis.

Canonical Pathway	*p*
Role of Oct4 in Mammalian Embryonic Stem Cell Pluripotency	0.0004
Pathogeneses of Multiple Sclerosis	0.0007
Human Embryonic Stem Cell Pluripotency	0.002
Role of NANOG in Mammalian Embryonic Stem Cell Pluripotency	0.002
Axonal Guidance Signaling	0.005
Aspartate Degradation II	0.009
IL-17A Signaling in Airway Cells	0.01
Virus Entry via Entry via Endocytic Pathways	0.01
Gα12/13 Signaling	0.01

**Table 3 nutrients-17-00921-t003:** OTUs that are more common in the IBS cohort than in HCs. Cases where the OTU is always or never present in a group are marked by an asterisk (*). Only substantial differences of 25% or more between the two groups (IBS and HC) were included.

OTUs Common in IBS but Uncommon in HC							
Phylum	Family	Genus	Species	IBS (n = 18)	IBS-D (n = 7)	IBS-C (n = 9)	HC (n = 18)
Bacteroidetes	Flavobacteriaceae	unclassified	unclassified	13	3	7	8
Bacteroidetes	Prevotellaceae	Prevotella	unclassified	6	2	2	1
Bacteroidetes	Prevotellaceae	Prevotella	unclassified	14	4	7	8
Bacteroidetes	Prevotellaceae	Prevotella	unclassified	15	5	7	9
Bacteroidetes	Rikenellaceae	unclassified	unclassified	6	2	4	1
Bacteroidetes	Rikenellaceae	unclassified	unclassified	15	4	9 *	9
Bacteroidetes	unclassified	unclassified	unclassified	10	2	6	5
Firmicutes	Lachnospiraceae	unclassified	unclassified	7	3	1	2
Firmicutes	Lachnospiraceae	unclassified	unclassified	9	3	5	2
Firmicutes	Lachnospiraceae	unclassified	unclassified	9	3	5	3
Firmicutes	Lachnospiraceae	unclassified	unclassified	10	2	6	2
Firmicutes	Lachnospiraceae	unclassified	unclassified	14	4	9	9
Firmicutes	Lachnospiraceae	unclassified	unclassified	14	5	7	9
Firmicutes	Ruminococcaceae	unclassified	unclassified	9	2	6	3
Firmicutes	Streptococcaceae	Streptococcus	lutetiensis	11	2	8	4
Firmicutes	unclassified	unclassified	unclassified	8	2	4	2
Firmicutes	unclassified	unclassified	unclassified	8	3	4	2
Fusobacteria	Fusobacteriaceae	Fusobacterium	unclassified	6	2	3	0 *
Fusobacteria	Fusobacteriaceae	Fusobacterium	unclassified	6	0 *	6	1
Proteobacteria	Alcaligenaceae	unclassified	unclassified	6	3	2	1
Proteobacteria	Chromatiaceae	unclassified	unclassified	9	3	6	4
Proteobacteria	Desulfobacteraceae	unclassified	unclassified	15	5	9 *	8
Proteobacteria	Pseudomonadaceae	Pseudomonas	unclassified	6	2	3	1
Proteobacteria	unclassified	unclassified	unclassified	6	1	3	1
Proteobacteria	unclassified	unclassified	unclassified	9	3	5	2
Proteobacteria	unclassified	unclassified	unclassified	9	4	3	3
Spirochaetes	Spirochaetaceae	Treponema	unclassified	8	1	4	3

**Table 4 nutrients-17-00921-t004:** OTUs that are more common in HCs than in the IBS cohort and IBS-subgroups. Cases where the OTU is always or never present in a group are marked by an asterisk (*). Only substantial differences of 25% or more between the two groups (IBS and HC) were included.

OTUs Common in HC but Uncommon in IBS							
Phylum	Family	Genus	Species	IBS (n = 18)	IBS-D (n = 7)	IBS-C (n = 9)	HC (n = 18)
Actinobacteria	Coriobacteriaceae	unclassified	unclassified	1	1	0 *	6
Actinobacteria	Micrococcaceae	Arthrobacter	unclassified	3	1	2	8
Actinobacteria	Streptomycetaceae	Streptomyces	unclassified	8	1	6	13
Bacteroidetes	Flavobacteriaceae	Capnocytophaga	ochracea	6	3	3	13
Bacteroidetes	Porphyromonadaceae	Porphyromonas	unclassified	3	1	2	9
Bacteroidetes	Rikenellaceae	unclassified	unclassified	7	3	2	15
Bacteroidetes	unclassified	unclassified	unclassified	5	1	4	10
Firmicutes	Bacillaceae	Bacillus	unclassified	4	1	3	9
Firmicutes	Bacillaceae	Bacillus	unclassified	6	2	4	11
Firmicutes	Lachnospiraceae	unclassified	unclassified	0 *	0 *	0 *	5
Firmicutes	Lachnospiraceae	unclassified	unclassified	0 *	0 *	0 *	5
Firmicutes	Lachnospiraceae	unclassified	unclassified	0 *	0 *	0 *	6
Firmicutes	Lachnospiraceae	unclassified	unclassified	1	0 *	1	6
Firmicutes	Lachnospiraceae	unclassified	unclassified	1	1	0	6
Firmicutes	Lachnospiraceae	unclassified	unclassified	1	0 *	1	6
Firmicutes	Lachnospiraceae	unclassified	unclassified	2	2	0 *	7
Firmicutes	Lachnospiraceae	Coprococcus	unclassified	2	1	1	8
Firmicutes	Lachnospiraceae	unclassified	unclassified	3	1	2	9
Firmicutes	Lachnospiraceae	unclassified	unclassified	3	0 *	3	9
Firmicutes	Lachnospiraceae	unclassified	unclassified	3	2	1	10
Firmicutes	Lachnospiraceae	unclassified	unclassified	4	1	3	10
Firmicutes	Lachnospiraceae	unclassified	unclassified	6	4	1	11
Firmicutes	Lachnospiraceae	unclassified	unclassified	4	0 *	4	12
Firmicutes	Lachnospiraceae	unclassified	unclassified	8	3	5	13
Firmicutes	Lachnospiraceae	unclassified	unclassified	9	3	5	14
Firmicutes	Lachnospiraceae	unclassified	unclassified	13	4	8	18 *
Firmicutes	Streptococcaceae	Streptococcus	unclassified	3	1	2	8
Firmicutes	Streptococcaceae	Streptococcus	gordonii	10	4	6	15
Firmicutes	unclassified	unclassified	unclassified	3	1	2	8
Firmicutes	Veillonellaceae	Dialister	invisus	11	4	5	16
Proteobacteria	Burkholderiaceae	unclassified	unclassified	3	2	1	8
Proteobacteria	Campylobacteraceae	Campylobacter	unclassified	5	1	4	11
Proteobacteria	Neisseriaceae	Neisseria	unclassified	6	1	5	11
Proteobacteria	Pasteurellaceae	Haemophilus	parainfluenzae	3	1	2	9
Proteobacteria	Pseudomonadaceae	Pseudomonas	unclassified	2	1	1	7
Proteobacteria	Pseudomonadaceae	Pseudomonas	unclassified	5	2	3	10
Proteobacteria	Pseudomonadaceae	Pseudomonas	unclassified	7	2	4	12
Proteobacteria	Pseudomonadaceae	Pseudomonas	unclassified	13	3	9 *	18 *
Proteobacteria	unclassified	unclassified	unclassified	1	0 *	1	7
Verrucomicrobia	unclassified	unclassified	unclassified	0 *	0 *	0 *	5
WS3	PRR-10	unclassified	unclassified	9	2	7	16

## Data Availability

The sequencing data that support the findings of this study are openly available in NCBI Gene Expression Omnibus at https://www.ncbi.nlm.nih.gov/geo, reference number GSE109597. Deposition date 24 January 2018. Last updated 18 June 2019.

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
