# Peer review of "Host Transcriptome and Microbial Variation in Relation to Visceral Hyperalgesia†"

_nutrients, 2025, doi:10.3390/nu17050921_

Round 1

Reviewer 1 Report

Comments and Suggestions for Authors

The manuscript presents an interesting multi-omics study investigating the relationships between the oral microbiome, peripheral blood transcriptome, and induced visceral pain (IVP) in healthy controls and IBS patients. The study is generally well-conceived, but several issues need addressing to strengthen the findings and conclusions.

  1. The cross-sectional design precludes causal inference. Emphasize that observed associations do not establish directionality.
  2. The assumption that the oral microbiome reflects gut microbiome alterations needs further justification and explicit acknowledgement of limitations.
  3. Confirm data distribution assessment (e.g., normality) and justify test selection (parametric vs. non-parametric).
  4. Clarify handling of the IBS-M subgroup (n=2). Consider exclusion or separate, cautious interpretation.
  5. The small n may limit the statistical power. Acknowledge the possibility of both type 1 and 2 errors.
  6. Detail control for potential confounders (i.e., diet, medication) and acknowledge any residual confounding effects.
  7. In addition to what is in the manuscript, add explicit comments regarding: lack of gut microbiome measurement, lack of functional assays, and limited sample size.
  8. Check for consistency in terminology (e.g., "OTU" vs. "taxa").

Author Response

Reviewer 1: The manuscript presents an interesting multi-omics study investigating the relationships between the oral microbiome, peripheral blood transcriptome, and induced visceral pain (IVP) in healthy controls and IBS patients. The study is generally well-conceived, but several issues need addressing to strengthen the findings and conclusions.

  1. The cross-sectional design precludes causal inference. Emphasize that observed associations do not establish directionality.

Thank you for the recommendation. We have included language highlighting the study design and need for caution in generalizing the data. Please see discussion section lines 518-523.

  1. The assumption that the oral microbiome reflects gut microbiome alterations needs further justification and explicit acknowledgement of limitations.

This is an excellent point. We have edited the introduction section and conclusion to reflect the uncertainties in the link between these two populations, please see the highlighted text.

  1. Confirm data distribution assessment (e.g., normality) and justify test selection (parametric vs. non-parametric).

Data exploration and handling was described in previous methodology from our lab. The relevant citation (number 43: Fourie, N.H., et al., The microbiome of the oral mucosa in irritable bowel syndrome. Gut Microbes, 2016. 7(4): p. 286-301.) has been added to the manuscript.

  1. Clarify handling of the IBS-M subgroup (n=2). Consider exclusion or separate, cautious interpretation.

Yes, agreed. These data were excluded when gene correlation was examined, please see line 457.

  1. The small n may limit the statistical power. Acknowledge the possibility of both type 1 and 2 errors.

Small sample size is addressed as part of limitations discussion in our discussion (highlighted text 618-629). While there are limited number of participants, the bacterial compositions and sequencing data are run in replicant and this provides additional statistical rigor that we hope you find acceptable.

  1. Detail control for potential confounders (i.e., diet, medication) and acknowledge any residual confounding effects.

We agree and have addressed this point as part of inclusion and exclusion criteria. Persons taking daily medications for gastrointestinal symptoms were not included in the study. Specific lifestyles (excessive caffeine and alcohol consumption, night shift work) and medications/supplements (laxatives, serotonin interfering agents, tricyclic antidepressants) were excluded. The effect of specific diet was less significant since participants were assessed on their response to our test solution.

  1. In addition to what is in the manuscript, add explicit comments regarding: lack of gut microbiome measurement, lack of functional assays, and limited sample size.

Comments have been added as recommended and noted in highlighted text 623-629.

  1. Check for consistency in terminology (e.g., "OTU" vs. "taxa").

Only OTU will be used, taxa has been edited out as recommended.

Reviewer 2 Report

Comments and Suggestions for Authors

This study, titled 'Host Transcriptome and Microbial Variation in Relation to Visceral Hyperalgesia' by Christopher J. Costa et al., examines the relationship between the circulating mRNA transcriptome, the IVP, and variations in the oral microbiome among participants. The research presents findings of interest and demonstrates sound methodology; however, certain aspects would benefit from revisions as follows:

  1. The background section of the abstract would benefit from a refocus to include the oral microbiome, as this plays a central role in the aim of the study.
  2. The manuscript contains an excessive number of figures. It is recommended that the authors consolidate the figures to produce more comprehensive multi-panel figures, ideally within the range of 4-5 total figures.
  3. In the introduction section where the authors mention about the systemic impact of dysbiosis in the oral microbiome, it would be interesting to cite recent studies that have explored the impact of oral health and the oral microbiota on cognitive decay: PMC11773611; PMC11760870.
  4. It is requested to expand the diagnostic criteria for IBS in the participants section of the materials and methods.
  5. Please, include the approval reference of the study.
  6. Kindly expand sections 2.3 and 2.4 to provide a replicable version of the methods employed, or reference previously utilized protocols.
  7. In the statistics section of the methods, specify the type of assessment conducted to evaluate the parametric or non-parametric nature of the obtained results, and describe the statistical analyses performed, as well as the chosen significance level. Additionally, provide details of the correlation analyses conducted.
  8. It remains unclear whether the following criteria were selected based on established guidelines or were arbitrarily chosen:‘To focus on meaningful changes in bacterial representation, the 187 analysis was limited to OTU with a substantial presence (≥30% of samples) within the 188 group under consideration (i.e., the abundance of an OTU present in 30% of HC sam-189 ples was compared to its abundance in IBS samples), which helps avoid outlier effect 190 of a single sample on the group’s microbiota composition. Abundances were consid-191 ered significant if there was a greater than 25% difference in presence between HC 192 and IBS groups…’ It is imperative to provide a clear justification for this aspect, as the obtained results are predicated on these criteria and require proper substantiation. Alternatively, if such justification cannot be provided, it is necessary to calculate coefficients of covariation or relevant thresholds based on established statistical criteria.
  9. The correlations presented in Figure 3 cannot be adequately associated with each specific number on the x-axis. A different type of graphical representation based on correlation points would likely be more suitable for visually comprehending the magnitude of the coefficient.
  10. Can figure 8 include the error bars? If not, please justify.
  11. It would be beneficial to provide a clear dysbiotic profile associated with hyperalgesia, which could inform potential therapeutic interventions based on the findings of the study. This aspect may be appropriately addressed within the discussion section of the article.

Author Response

Reviewer 2: This study, titled 'Host Transcriptome and Microbial Variation in Relation to Visceral Hyperalgesia' by Christopher J. Costa et al., examines the relationship between the circulating mRNA transcriptome, the IVP, and variations in the oral microbiome among participants. The research presents findings of interest and demonstrates sound methodology; however, certain aspects would benefit from revisions as follows:

  1. The background section of the abstract would benefit from a refocus to include the oral microbiome, as this plays a central role in the aim of the study.

Thank you for your suggestion. The introduction has been edited to focus more on the oral microbiome. Please see the highlighted text lines 83-100 as well as 105-110.

  1. The manuscript contains an excessive number of figures. It is recommended that the authors consolidate the figures to produce more comprehensive multi-panel figures, ideally within the range of 4-5 total figures.

We have grouped related figures into multi-tiled figures and have moved some of the figures to the supplement section. Please see updated manuscript and supplemental figures based on your suggestion.

  1. In the introduction section where the authors mention about the systemic impact of dysbiosis in the oral microbiome, it would be interesting to cite recent studies that have explored the impact of oral health and the oral microbiota on cognitive decay: PMC11773611; PMC11760870.

Thank you for bringing these emerging studies to our attention. We have incorporated them into the introduction section, please see highlighted text in lines 105-110.

  1. It is requested to expand the diagnostic criteria for IBS in the participants section of the materials and methods.

Patients were diagnosed according the ROME III criteria for IBS, highlighted line 148-149. Additional significant inclusion and exclusion criteria are included. The full list of inclusion and exclusion criteria is available on clinicaltrials.gov using the referenced clinical trial number.

  1. Please, include the approval reference of the study.

Approval reference has been added, line 152.

  1. Kindly expand sections 2.3 and 2.4 to provide a replicable version of the methods employed, or reference previously utilized protocols.

Sections 2.3 and 2.4 have been expanded to include pertinent details as recommended.

  1. In the statistics section of the methods, specify the type of assessment conducted to evaluate the parametric or non-parametric nature of the obtained results, and describe the statistical analyses performed, as well as the chosen significance level. Additionally, provide details of the correlation analyses conducted.

Methods were developed based on prior analysis conducted by our group. The relevant citation with full methods discussion has been included, line 262-264.

  1. It remains unclear whether the following criteria were selected based on established guidelines or were arbitrarily chosen: ‘To focus on meaningful changes in bacterial representation, the 187 analysis was limited to OTU with a substantial presence (≥30% of samples) within the 188 group under consideration (i.e., the abundance of an OTU present in 30% of HC sam-189 ples was compared to its abundance in IBS samples), which helps avoid outlier effect 190 of a single sample on the group’s microbiota composition. Abundances were consid-191 ered significant if there was a greater than 25% difference in presence between HC 192 and IBS groups…’ It is imperative to provide a clear justification for this aspect, as the obtained results are predicated on these criteria and require proper substantiation. Alternatively, if such justification cannot be provided, it is necessary to calculate coefficients of covariation or relevant thresholds based on established statistical criteria.

Methods were developed based on prior analysis conducted by our group. The relevant citation with full methods discussion has been included, line 262-264.

  1. The correlations presented in Figure 3 cannot be adequately associated with each specific number on the x-axis. A different type of graphical representation based on correlation points would likely be more suitable for visually comprehending the magnitude of the coefficient.

This figure has been moved to the supplement and relabeled Supplemental Figure 1A. We have attempted and the addition of x-axis line markers would make the image too cluttered and hard to read/comprehend. Further discussion of magnitude of coefficient is present in the text of the article.

  1. Can figure 8 include the error bars? If not, please justify.
    1. This figure has been moved to the supplement and re-labeled as Supplemental Figure 1B. It is in the author’s opinion that error bars in this figure are not necessary given that it is a visual representation of data present in Tables 3 & 4 and Supplemental Tables S4 and S5. We hope that you find the justification satisfactory as the recommended revision is not practicable at this time.
  2. It would be beneficial to provide a clear dysbiotic profile associated with hyperalgesia, which could inform potential therapeutic interventions based on the findings of the study. This aspect may be appropriately addressed within the discussion section of the article.

Thank you. We have made added comments in discussion section, lines 559-590. Additionally, summary statement has been added and highlighted at lines 629-631.

Round 2

Reviewer 1 Report

Comments and Suggestions for Authors

Good to go

Comments on the Quality of English Language

Not bad

Reviewer 2 Report

Comments and Suggestions for Authors

The authors have stisfactorily addressed my previous concerns in the revisons performed, and the manuscript has been substantially improved.